# How does CEO narcissism affect enterprise ambidextrous technological innovation? The mediating role of corporate social responsibility

**Zeping Wang** [ORCID]*, **Xingqiu Hu, Feifei Yu**

School of Business, Hohai University, Nanjing, China

* 1625288169@qq.com

## Abstract

In the context of a dynamic environment and increasing competition, innovation is the key for companies to gain long-term growth. And narcissism, as an important psychological factor influencing CEOs to make corporate decisions, has a significant impact on corporate innovation strategies. This study explores localized dimensions and ways of measuring narcissism among Chinese CEOs. Based on the upper echelons theory, using data from R&D-intensive firms listed in Shanghai and Shenzhen A-shares from 2015–2020, this study empirically examines the effect of CEO narcissism on exploratory and exploitative innovation and the mediating role of corporate social responsibility. The results show that: CEO narcissism has a positive effect on corporate ambidextrous technological innovation and a more significant effect on exploratory innovation; the mediating role of corporate social responsibility is all verified. These findings provide a reference for listed companies to select and hire CEOs scientifically and rationally, and have important implications for companies to develop long-term innovation strategies.

## 1 Introduction

In a constantly changing and competitive business environment, it is a great challenge for companies to realize the growing demands from the market and customers. One of the themes of innovation research is how companies learn and produce knowledge to survive and compete in an increasingly dynamic environment. Therefore, innovation is a key element to ensure the competitiveness and success of organizations [1].

Upper echelons theory holds that "organizations are a reflection of the personal characteristics of top managers" [2]. Previous studies were mostly based on the theory of objective rationality, focusing on the rational economic man hypothesis to analyze the relationship between CEO demographic characteristics (such as gender, education, age, tenure, employment experience, professional background, etc.) and corporate behavior. With the emergence of the theory of bounded rationality and the theory of social cognition, more and more studies have confirmed that the CEO, as the main body of corporate decision-making, cannot be completely

**Funding:** This work was supported by the general project of the National Social Science Foundation of China (20BGL008) and the Fundamental Research Funds for the Central Universities (B210207026). The funders had no role in study design, data collection and analysis, decision to publish, or preparation of the manuscript.

**Competing interests:** The authors have declared that no competing interests exist.

rational [3]; the psychological traits of CEOs can affect their decision-making and thus have an impact on the enterprise [4]. In this context, narcissism as a psychological trait that reflects an individual's high self-confidence and desire for attention and praise has become a defining characteristic of some CEOs [5], and the role played by CEO narcissism in corporate decision making, especially in corporate innovation-driven strategies, has received increasing attention.

It has been shown that narcissistic CEOs tend to invest in new technologies, which to some extent promotes continuous innovation in the firm. The reason behind this is that the narcissistic CEO's supreme self-confidence can reduce the impact of uncertainty and risk on the CEO's decision making, thus enhancing his decision speed and finally actively investing in new technologies [6]. Gerstner (2013) stated that the higher the narcissistic tendency of managers, then the lower their avoidance motivation and the more likely they are to overestimate the potential benefits of risk and enhance the firm's adoption of technological discontinuous innovations [7]. Wales et al. (2013) stated that narcissistic CEOs may lead firms to take bolder and more aggressive strategic actions and propose ambitious innovation visions, thus CEO narcissism is positively related to entrepreneurial orientation, although entrepreneurial orientation has the potential to generate negative returns [8]. Kashmiri et al. (2017) studied the effect of CEO narcissism on technological innovation, stating that because narcissistic CEOs believe too much in their own abilities, they may be optimistic about the success of new products, thus promoting the introduction of new products and increasing the speed of technological innovation [9]. Du et al. (2018) pointed out that the trait that managers' narcissism likes to be praised and even touted by others will not only significantly promote corporate employees to make suggestions, but also enhance subordinates' innovation consciousness and stimulate their innovative energy, thus CEO narcissism is conducive to the cultivation and development of corporate innovation climate [10].

It can be seen that although existing studies confirm the positive impact of CEO narcissism on firm innovation, they mostly ignore the heterogeneity of firm innovation. March (1991) proposed the concept of dual innovation, arguing that both exploration-based and exploitation-based innovation are essential for firms, and innovation for both purposes is dual innovation [11], and they have different effects on firm behavior [12]. However, existing studies have not discussed clearly the impact of CEO narcissism on dual innovation in firms. To fill this research gap, this study divides innovation into exploratory and exploitative innovation and investigates the relationship between CEO narcissism and them separately. In addition, most previous studies have explored the direct impact of narcissistic traits on corporate innovation and lacked a discussion of the pathways of the role of CEO narcissism in influencing innovation. In this paper, in the course of studying CEO narcissism and corporate innovation, the role of corporate social responsibility between CEO narcissism and corporate innovation gradually comes into focus, and it seems to play a linking role between CEO narcissism and corporate innovation. On the one hand, it has been demonstrated that CEO narcissism can promote corporate social responsibility. Tang et al. (2018) argued that narcissistic managers have strong charisma and quality leadership, which can motivate companies to take more social responsibility, thus enabling them to establish a quality philanthropic image in the social environment and ultimately achieve the goal of enhancing their personal reputation [13]. Petrenko et al. (2016) argue that CEOs can significantly enhance corporate social responsibility out of a desire for attention [14]. On the other hand, the fulfillment of CSR can provide resources for corporate innovation. Corporate investment in CSR allows companies to build a deeper and broader network of relationships with stakeholders, prompting them to share and exchange external information with stakeholders and use the acquired external information to improve internal information, thus improving the company's innovation capacity [15,16]. Therefore, it is

necessary to include CSR in the research framework of the relationship between CEO narcissism and firms' dual technological innovation and to explore whether CSR has a mediating effect.

In summary, this study selects R&D-intensive firms listed in A-share in Shanghai and Shenzhen from 2015 to 2020 as a sample according to the Classification Guidelines for Listed Companies of China Securities Regulatory Commission to study the impact of CEO narcissism on corporate dual technological innovation and analyze the mediating role of corporate social responsibility between CEO narcissism and corporate dual technological innovation.

The possible theoretical contributions of this paper are as follows: (1) Most of the existing research focuses on the influence of CEO demographic characteristics on corporate behavior, but not enough attention is paid to CEO psychology and personality traits. This paper discusses the personality trait of CEO narcissism, enriches and deepens the related research on the upper echelon theory; (2) Most of the existing studies on CEO narcissism traits have been discussed based on Western contexts, but there are differences between Chinese and Western cultures, and the measurement dimensions adopted in the West are not necessarily applicable to China. Therefore, this paper discusses and adopts suitable non-interventional indicators for measuring CEO narcissism based on Western research and in the context of China. This finding may provide a reference for future measurement of CEO narcissism. (3) Although existing studies have discussed the role of different CEO types on corporate technological innovation, most of them regard corporate innovation behavior as homogeneous behavior, ignoring the heterogeneity of corporate innovation. This paper starts from the theory of ambidextrous innovation, studies the influence of CEO characteristics on the technological innovation of enterprises in different dimensions, and provides new research ideas for the contradictions and differences in the existing research conclusions. (4) This paper investigates the role of corporate social responsibility in the relationship between CEO narcissism and corporate innovation, enriching the literature on the field of CEO narcissism and corporate innovation.

## 2 Theoretical analysis and research hypothesis

### 2.1 CEO narcissism and enterprise ambidextrous technological innovation

Traditional economics research is based on the assumption of "rational economic man", which assumes that executives have all the information and knowledge about the state of the economic environment in all aspects of the enterprise. However, in real life, people cannot be infinitely rational. Due to the complexity of the internal and external environment of enterprises and the limitation of limited rationality of executives, it is impossible for executives to know all the information about the internal and external environment of enterprises. Therefore, executives can only make strategic decisions based on their own knowledge framework and cognitive structure, which ultimately have an impact on the organization. Based on the research related to limited rationality, Hambrick and Mason (1984) proposed the upper echelon theory [2]. In the analysis of the upper echelon theoretical framework, the operation of a firm is the result of the entire executive team working together. When studying corporate actions, top managers should be the focus of the study, and the impact of different traits of managers on corporate strategy formulation and strategy implementation should be studied according to the differences in their own traits and position power.

With the gradual increase in the number of individuals with narcissistic personality traits in organizations, scholars in the field of management and organizational behavior have begun to include narcissism as a personality trait in studies related to CEO and corporate behavior. Narcissism is a personality trait, a unique and superior perception of self [17]. It not only symbolizes the degree of self-centeredness and self-inflation of the individual, but also implies that the

individual continues to seek attention and admiration from others, while craving praise, honor, and power [18,19]. With the gradual deepening of research on narcissistic personality traits, scholars in the fields of management and organizational behavior began to incorporate narcissism as a personality trait into the research on CEO and corporate behavior. Maccoby (2017) pointed out that narcissists are self-centered and eager to gain admiration from others, which makes narcissists tend to have better expectations for results, are more willing to try challenges, and can overcome difficulties and achieve goals [20]. Therefore, narcissists often become managers in the organization. Rosenthal and Pittinsky (2006) formally introduced managerial narcissism into management theory research for the first time, and systematically expounded the connotation and influence of managerial narcissism. They pointed out that managers' decision-making motivation is not only driven by the interests of the enterprises they manage, but also driven by bounded rational personal needs and ideas. Therefore, CEO narcissism may have an important impact on corporate innovation activities [21].

Based on the dual innovation theory, corporate innovation can be divided into two parts: exploratory and exploitative innovation [11]. Exploratory innovation is an invention based on a different set of technological principles, and exploitative innovation is an improvement and extension of existing capabilities, technologies and paradigms. Both are important strategic tools for firms to develop technological innovation and seek sustainable development [22], and there are significant differences between different types of technological innovation in terms of resource investment, implementation risks, and outcome benefits [23]. Therefore, it is necessary to analyze the impact of CEO narcissism on innovation from these two components separately.

(1) CEO Narcissism and Exploratory Innovation

Exploratory innovation is based on different technical principles than the original product, usually requires long-term investment, and is relatively risky [24]. Although exploratory innovation has more uncertainty and investment risks, the success of exploratory innovation can often open up new markets [25]. Narcissistic CEOs have high self-confidence in their abilities and think they are superior to others. This supreme self-confidence tends to make them overestimate the potential benefits of risk [7]. Narcissistic CEOs believe they can lead the company to innovation better than others, so they tend to invest in new technologies, promote fundamental innovation [8], and provide development impetus for enterprise exploratory innovation. Secondly, with the development of science and technology, innovation ability has become the core element of enterprise technology competition. The results of exploratory innovation have breakthroughs, are more likely to receive media coverage and social attention, and are more likely to be loved and favored by consumers when launched. Under the influence of narcissistic traits, external attention is important for narcissistic CEOs to achieve self-satisfaction. They need to show their superiority through external recognition and rewards [26]. Therefore, exploratory innovation can help narcissistic CEOs attract the media's and the public's attention and maintain their superiority. Under this incentive, narcissistic CEOs will likely support exploratory innovation in corporate development. As a result, the following hypothesis is proposed.

H1a: CEO narcissism can promote enterprise exploratory innovation.

(2) CEO Narcissism and Exploitative Innovation

Exploitative innovation usually utilizes or upgrades existing resources and products, and the changes are relatively small. Therefore, compared with exploratory innovation, exploitative innovation has lower risks and costs. Although narcissistic CEOs have a higher level of risk-taking, they also control the scope of risk in decision-making under term constraints [27]. At the same time, when making corporate strategic decisions, CEOs will form thinking patterns,

management patterns, successful experiences, and job role dependencies based on knowledge background, personal cognition, and values [28], which has behavioral inertia.

Therefore, it is also possible to choose a relatively easier path of innovation. Second, although exploitative innovation cannot help companies enter new fields, its short innovation cycle can help companies consolidate their existing positions in time [29]. Narcissistic CEOs may also develop exploitative innovations to demonstrate competence and achieve a sense of superiority over others during their tenure. In addition, narcissistic CEOs may lead companies to take more active strategic actions and put forward a grand vision for innovation, which also helps to enhance employees' awareness of innovation and cultivate an atmosphere of innovation in the enterprise [20] so that e employees can actively access and use internal and external information, ultimately facilitating the development of developmental innovation. As a result, the following hypothesis is proposed.

H1b: CEO narcissism can promote enterprise exploitative innovation.

## 2.2 The mediating role of corporate social responsibility

(1) CEO Narcissism and Corporate Social Responsibility

According to signaling theory, fulfilling CSR enables companies to send positive signals to stakeholders that they are operating with integrity and responsibility, which in turn creates a positive corporate image as well as increases public attention to the company and its top executives. Narcissistic CEOs prefer to show their ego, and fulfilling CSR can bring them more attention and maintain their superiority psychology, thus they will support external CSR fulfillment [30,31]. In addition, narcissistic CEOs have high identification with their personal capabilities and high expectations about the profitability of the companies they manage [32] and will tend to increase their commitment to CSR fulfillment.

(2) Corporate Social Responsibility and Exploratory Innovation

Based on stakeholder theory, Freeman and Reed (1983) categorized stakeholders into 3 categories, ownership stakeholders, economically dependent stakeholders, and social stakeholders. These 3 types of stakeholders mainly include corporate shareholders, creditors, employees, suppliers, consumers, government, and public welfare organizations [33]. Companies that fulfill a high level of social responsibility pay more attention to all stakeholders while focusing on the interests of shareholders [34]. This means that the fulfillment of social responsibility can expand the scope of communication between enterprises and stakeholders, which in turn helps enterprises obtain more diverse and heterogeneous resources: through social responsibility, enterprises can often build a good reputation and prestige, which brings them moral capital and improves their credit level, thus reducing the risk identification of creditors and obtaining recognition in the capital market; social responsibility is often seen as a rent-seeking way to establish political relationships and obtain political resources. These heterogeneous resources provide support for improving the level of technical attributes and legal attributes of enterprise patent quality, which in turn guarantees the output of high-level patents and enhances the level of enterprise technological innovation.

As Brown and Forster (2013) pointed out, companies that proactively fulfill their social responsibilities can establish a wider range of stakeholder relationships than companies that simply fulfill their responsibilities clearly defined by laws and regulations [35]. Extensive stakeholder relationships can help companies keep abreast of market needs, understand the necessary information and other effective advice needed to develop new products [36], and provide external heterogeneous knowledge sources for enterprises to help them enter new knowledge fields and realize the development of exploratory innovation. At the same time, enterprises'

sunk cost in developing exploratory innovation is reduced as much as possible with the support of internal and external resources.

As a result, the following hypothesis is proposed.

H2a: Corporate social responsibility mediates CEO narcissism and exploratory technological innovation.

(3) Corporate Social Responsibility and Utilization-based Innovation

According to stakeholder theory, companies that fulfill their social responsibility are able to further strengthen and develop partnerships with their existing stakeholders, thus gaining positive feedback and effective suggestions from stakeholders such as employees, consumers, business partners, financial institutions, and government departments [37]. These suggestions can support corporate innovation activities. Feedback from business partners can further explore the potential resources in the existing technical knowledge system, which can be integrated and analyzed to standardize and refine the current technology, and also use these sources to effectively modify and upgrade products. Optimization suggestions for existing products from consumers, production partners, and others lead to refinement of categories, quality improvement, and production flexibility [12]. These changes have a positive effect on firms' exploitative innovation.

As a result, the following hypothesis is proposed.

H2b: Corporate social responsibility mediates CEO narcissism and exploitative technological innovation.

Based on the analysis above, the research hypothesis model of this paper is constructed (Fig 1).

## 3 Research design

### 3.1 Sample selection and data sources

Research shows that pharmaceutical manufacturing, general equipment manufacturing, special equipment manufacturing, automobile manuf sacturing, electrical manufacturing,

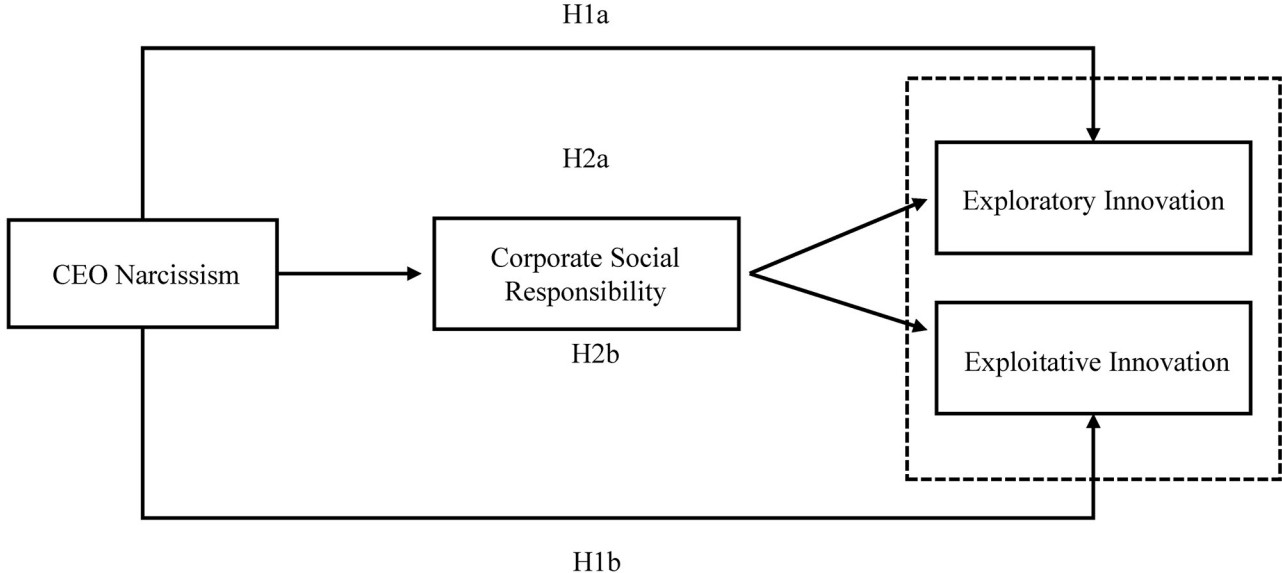

**Fig 1. Research hypothesis mode.**

electronics manufacturing, and information technology service industries are R&D-intensive industries, and the technological innovation activities of their enterprises are relatively active [38], which is suitable for the analysis of enterprise technological innovation. According to the enterprise classification of R&D-intensive industries in the "Guidelines for the Classification of Listed Companies" of the China Securities Regulatory Commission, combined with the availability of CEO narcissism data, this paper selects the companies listed on the Shanghai Stock Exchange and Shenzhen Stock Exchange from 2015 to 2020 as observation samples. And other select listed companies that meet the following conditions for research: (1) The starting date of the CEO's tenure is at least 2015 or before; (2) The CEO's tenure year should be within the time range of this research, and the CEO has worked in the company for at least four years; (3) The basic data required for research can be obtained: data required to measure the degree of narcissism (company news, character column reports, CEO personal speeches, etc.), data to measure the company's dual technological innovation, corporate social responsibility score, and other variables required data.

In the data of this article, the ratio of the number of news stories with the CEO's name in the news headline to the total number of news for the entire statistical year and the ratio of the number of singular first-person pronouns used by the CEO to the sum of all first-person pronouns in public speeches or interviews are all collected manually from the official websites of sample companies, annual reports, China Finance and Economics website, etc. CSR data comes from Hexun.com, and other data comes from CSMAR database and CNRDS database.

## 3.2 Definition and measurement of variables

**3.2.1 Dependent variable.** Scholars have different opinions about the classification method of dual innovation. Christensen and Bower (1996) classified innovation into disruptive and sustaining innovation based on whether the innovation disrupts existing technology [39], He and Wong (2004) classified innovation into exploratory and exploitative innovation based on the degree of newness of the product market segment for which the innovation activity is oriented [22], and Freeman and Soete (2009) argue that innovation can be classified as breakthrough and incremental innovation depending on the degree of innovation [40]. Although there are various statements on the classification of dual innovation, they can all be seen as innovation based on exploration and exploitation in essence. Studies have shown that invention patent is a concept that is more in line with exploratory innovation in terms of product development in new markets and achieving technological breakthroughs, and can be used to reflect the results of exploratory innovation in enterprises; utility model and design patents mainly focus on improving on the original technology, which is an extension of existing products and technologies, and can be used to reflect the results of exploitative innovation in enterprises [41–43]. Therefore, we select the authorized amount of enterprise invention patents to represent exploratory innovation and the authorized amount of enterprise appearance and utility model patents to measure exploitative innovation.

**3.2.2 Independent variable.** Measuring the level of narcissism in CEOs is a key aspect of studying CEO narcissism. Questionnaire measures of narcissism are well established in the academic community. The most widely used is based on the Narcissistic Personality Inventory (NPI) developed by Raskin and Hall in 1979, which gradually evolved into the 16-item Narcissistic Personality Inventory (NPI-16). With the ease of access to media materials, Petrenko et al. (2016) [14] and Fung et al. (2020) [44] adopted a new video survey method based on the NPI-16 scale and hired a third-party rater to measure the narcissism level of managers. In the actual study, it is difficult for scholars to systematically concentrate on questionnaires based on the NPI for executives of listed companies. And executives of listed companies often do not

tend to participate in surveys involving personal personality traits, especially controversial and sensitive issues such as narcissism [45]. Webb and Weick (1979) have also stated that social science research should observe traces left by people in the physical environment, non-participant observations, and written and speaking habits as a way of understanding respondents' preferences, opinions, and traits; a non-positive survey approach can reduce problems such as investigator reactivity, demand characteristics, and bias brought about by the researcher's subjective expectations [46]. Therefore, Chatterjee and Hambrick (2007) pioneered a novel measure of managerial narcissism, the non-interventional measure [47], which operates by browsing objective information related to managers to obtain proxies that measure their narcissism, which are called non-interventional indicators. In the article, they designed five non-intervention indicators, namely: (i) the size of the manager's photo in the firm's annual report; (ii) the manager's standing position in the firm's external presentation; (iii) the frequency of the manager's use of the first person in personal speeches or interviews; (iv) the pay gap between the manager and the second ranked executive; (v) the gap in the firm's non-cash benefits between the manager and the second ranked executive. This is not very different from the four dimensions of narcissism (authority, self-attraction, self-superiority, and desire for power) identified by Emmons (1987) using an exploratory factor approach through the NPI [17], which provides a good representation of the narcissistic behavior of corporate managers. This study not only qualitatively illustrates how non-interventional indicators fit into the four sub-dimensions of narcissistic personality, but also confirms through the data that they can be fitted into a narcissism index as a measure of managers' narcissistic tendencies. Therefore, this paper adopts a non-interventional measure. Drawing on the measure used by Chatterjee and Hambrick (2007) in their study of the relationship between CEO narcissism and corporate strategy and business performance, that is, using CEO photographs, C EO compensation, CEO cash, non-cash benefits and the frequency of CEOs using the first person in the interview, which are five objective indicators to measure the degree of CEO narcissism. Considering that there are fewer photos of company leaders displayed in the annual reports of Chinese listed companies, the indicator of CEO personal photo size and position in the company's annual report is thus discarded. CEO compensation and benefit packages in most Chinese state-owned enterprises are not determined by the CEOs themselves, but controlled by the SASAC, combined with the idea proposed by Ham (2018) that narcissistic CEOs enjoy higher relative compensation. Therefore, CEO pay is defined as the relative pay of the CEO to other employees as a measure [48]. Finally, the following three indicators are used to measure the narcissism level of corporate CEOs:

1. To examine the prominent position of the CEO in corporate press releases and to count the ratio of the number of news stories with the CEO's name in the news headlines to the total number of news stories in the entire statistical year;

2. The ratio of the number of singular first-person pronouns used by the CEO to the sum of all first-person pronouns in public speeches or interviews;

3. The ratio of the CEO's cash compensation to the average compensation of the top three executives in the company.

The relationship between the three indicators in this paper and each dimension of narcissism is shown in Table 1.

To further test the consistency of the 3 objective indicators, we performed validation factor analysis and reliability analysis using A-mos21.0 (CFI = 0.942, NFI = 0.941, IFI = 0.942) and the Cronbach's alpha value of the measure was 0.7, which is higher than the minimum acceptable value of 0.6, indicating that the CEO narcissism measurement index chosen for this article

**Table 1. The index of CEO narcissism and the four dimensions of narcissism.**

| | leadership/authority | Ego/Self Worship | superiority/arrogance | Utilitarianism/desire for power |
|---|---|---|---|---|
| **The four dimensions of narcissism (Emmons,1987)** | | | | |
| Classic items in the NPI | I was born to be in charge; others should obey my orders | I've always appreciated everything about myself; I'm a great person myself | I want to be the center of attention; I like to be above others | I must be more successful than others; I must be respected and revered by others |
| **The correspondence between indicator meaning and narcissism subdimensions** | | | | |
| The ratio of the number of times the CEO is reported to the total number of news on the company's official website news | I should be in control of corporate news coverage | I deserve more attention | I am the core of the business | People inside the business should know what's going on with me |
| The ratio of the first-person pronoun singular to the sum of the first-person pronouns (such as "I" and "we") used by CEOs in public speeches or interviews | Most of the achievements of the enterprise are brought by me | My thoughts and opinions are the most correct | My role is the most important to the business compared to others | My importance in the company needs to be highlighted |
| The ratio of CEO pay to the average pay of the top three executives | I am ahead of others in every way | I deserve the highest recognition for my work ability | I am the key to business results | I should be paid more than others |

has good reliability. Since narcissism is a psychological characteristic that does not easily change with external influences, it is relatively stable to infer the degree of narcissism in other periods by obtaining the degree of narcissism of the CEO in one period [5]. Therefore, this paper draws on the approach of Wen (2015) and Buyl et al. (2019) to determine the degree of narcissism of CEOs by selecting the method of averaging two years of data during the CEO's tenure [49,50].

**3.2.3 Mediating variable.** CSR data is obtained from Hexun Station's 2015–2020 CSR report rating database. Hexun Station is a more authoritative website that publishes comprehensive CSR scores of Chinese listed companies. Hexun Station publishes a comprehensive social responsibility rating of Chinese listed companies every year as a measure of CSR performance. It can reflect the social responsibility performance of enterprises in an objective manner.

**3.2.4 Control variable.** We refer to the selection of control variables in the relevant literature on CEO narcissism and ambidextrous innovation. We introduce CEO age (Age), CEO education (Edu), and two jobs (Post) as control variables from the CEO level. The company's asset-liability ratio (Debt), company scale (Size), company listing years (Set), company equity nature (Pattern), and company R&D investment (Rd) are introduced as control variables; at the same time, to control the industry and year fixed effects, we introduce Industry and year dummy variables. The specific measurement methods of variables are shown in Table 2.

### 3.3 Model construction

In order to test the research hypothesis of this paper, the following model was constructed in this study.

$$lnPat1_{it} = \beta_0 + \beta_1 Nar_{it} + \sum_1^m \alpha_m Control_m + \sum_1^k \lambda_k Industry_{it} + \sum_1^l \delta_i Year_{it} + \varepsilon_{it} \quad (1)$$

$$lnPat2_{it} = \beta_0 + \beta_1 Nar_{it} + \sum_1^m \alpha_m Control_m + \sum_1^k \lambda_k Industry_{it} + \sum_1^l \delta_i Year_{it} + \varepsilon_{it} \quad (2)$$

$$CSR_{it} = \beta_0 + \beta_1 Nar_{it} + \sum_1^m \alpha_m Control_m + \sum_1^k \lambda_k Industry_{it} + \sum_1^l \delta_i Year_{it} + \varepsilon_{it} \quad (3)$$

**Table 2. Variable definitions.**

| Variable category | Variable name | Variable symbol | Variable definitions |
|---|---|---|---|
| dependent variable | exploratory innovation | LnPat1 | Take the natural logarithm of (number of invention patents + 1) |
| | exploitative innovation | LnPat2 | Take the natural logarithm of (the number of utility models and appearance patents + 1) |
| independent variable | CEO Narcissism | Nar | CEO Narcissism Index |
| mediating variable | Corporate Social Responsibility | CSR | CSR Rating Score released by China Hexun.com |
| control variable | Enterprise asset-liability ratio | Debt | The ratio of the company's total liabilities to total assets at the end of the current period |
| | Enterprise size | Size | The natural logarithm of the total assets of the enterprise at the end of the current period |
| | Enterprise years | Set | Year of establishment |
| | The nature of corporate equity | Pattern | The nature of enterprise equity, 1 = state-owned, 0 = non-state-owned |
| | Enterprise R&D investment | Rd | Enterprise R&D investment, the ratio of R&D investment to operating income*100 |
| | CEO's age | Age | CEO's age |
| | CEO's education | Edu | CEO education, 1 = technical secondary school or below, 2 = junior college, 3 = undergraduate, 4 = master, 5 = doctoral, 6 = MBA/EMBA |
| | Is the CEO concurrently | Post | Whether the CEO is both the CEO and the general manager of the company, 1 = yes, 0 = no |
| | industry | Industry | Industry dummy variables, according to the 2012 industry classification of the China Securities Regulatory Commission |
| | years | Year | Year dummy variable |

$$lnPat1_{it} = \beta 0 + \beta 1 Nar_{it} + \beta 2 CSR_{it} + + \sum_1^m \alpha_m Control_m + \sum_1^k \lambda_k Industry_{it}$$
$$+ \sum_1^l \delta_i Year_{it} + \varepsilon_{it} \tag{4}$$

$$lnPat2_{it} = \beta 0 + \beta 1 Nar_{it} + \beta 2 CSR_{it} + + \sum_1^m \alpha_m Control_m + \sum_1^k \lambda_k Industry_{it}$$
$$+ \sum_1^l \delta_i Year_{it} + \varepsilon_{it} \tag{5}$$

lnPat1it and lnPat2it represent the exploratory innovation and utilization innovation of enterprise i in the t year, respectively. Nar is the CEO's narcissism level. Control represents a control variable, specifically the CEO background characteristics and company-related data described above. Industry and Year represent industry and year dummy variables, respectively, and εit is a random disturbance term.

Model 1 and Model 2 are used to examine the impact of CEO narcissism on exploratory innovation and exploitative innovation. Model 4 and Model 5 are used to test the mediating role of corporate social responsibility in CEO narcissism and corporate exploratory innovation and corporate exploitative innovation.

## 4 Empirical results and analysis

### 4.1 Descriptive statistics

Table 3 shows the descriptive statistics of the main variables, with a total of 789 samples. The mean values of exploratory and exploitative innovation are 1.704 and 2.39, respectively, indicating that exploitative innovation has relatively more achievements in R&D-intensive

**Table 3. Descriptive statistics for primary variables.**

| variable | N | mean | p50 | sd | min | max |
|---|---|---|---|---|---|---|
| Pat1 | 798 | 1.704 | 1.386 | 1.524 | 0 | 7.151 |
| Pat2 | 798 | 2.39 | 2.25 | 1.806 | 0 | 8.198 |
| Nar | 798 | 0.313 | 0.284 | 0.14 | 0.067 | 0.664 |
| CSR | 798 | 23.17 | 22.09 | 13.96 | -3.31 | 75.51 |
| Debt | 798 | 0.376 | 0.368 | 0.168 | 0.069 | 0.819 |
| Size | 798 | 22.37 | 22.17 | 1.113 | 20.58 | 26.25 |
| Set | 798 | 13.13 | 12 | 3.935 | 7 | 28 |
| Pattern | 798 | 0.18 | 0 | 0.385 | 0 | 1 |
| Rd | 796 | 6.972 | 5.1 | 5.641 | 1.09 | 40.06 |
| Age | 798 | 50.35 | 51 | 6.535 | 34 | 66 |
| Edu | 798 | 3.708 | 4 | 1.016 | 2 | 6 |
| Post | 798 | 0.358 | 0 | 0.48 | 0 | 1 |

enterprises. The maximum value of the CEO narcissism index is 0.664, the minimum value is 0.067, the median is 0.284, and the average value is 0.313, indicating that there is still a large gap in the narcissism index of CEOs of different companies, and CEOs with relatively low levels of narcissism occupy a large proportion of most.

## 4.2 Correlation analysis

In this paper, the Pearson correlation coefficient test was carried out on the relevant variables, and the specific correlation coefficient test results are shown in Table 4. According to the correlation coefficient results in Table 4, the correlation coefficient between exploratory innovation and CEO narcissism is 0.197, the correlation coefficient between exploratory innovation and corporate social responsibility is 0.252, the correlation coefficient between exploitative innovation and CEO narcissism is 0.133, the correlation coefficient between exploitative innovation and corporate social responsibility is 0.120, and the correlation coefficient between corporate social responsibility and CEO narcissism is 0.191. All results are significant. It shows a relationship between the variables selected in this paper, which can be further analyzed.

**Table 4. Correlation coefficient test of each variable.**

| Variable | Pat1 | Pat2 | Nar | CSR | Debt | Size | Set | Pattern | Rd | Age | Edu | Post |
|---|---|---|---|---|---|---|---|---|---|---|---|---|
| Pat1 | 1 | | | | | | | | | | | |
| Pat2 | 0.646*** | 1 | | | | | | | | | | |
| Nar | 0.197*** | 0.133*** | 1 | | | | | | | | | |
| CSR | 0.252*** | 0.120*** | 0.191*** | 1 | | | | | | | | |
| Debt | 0.276*** | 0.389*** | 0.173*** | -0.061* | 1 | | | | | | | |
| Size | 0.586*** | 0.498*** | 0.288*** | 0.273*** | 0.536*** | 1 | | | | | | |
| Set | 0.210*** | 0.130*** | 0.228*** | 0.104*** | 0.272*** | 0.333*** | 1 | | | | | |
| Pattern | 0.133*** | 0.045 | 0.150*** | 0.061* | 0.159*** | 0.115*** | 0.194*** | 1 | | | | |
| Rd | 0.158*** | -0.158*** | -0.132*** | 0.024 | -0.213*** | -0.098*** | -0.067* | 0.129*** | 1 | | | |
| Age | 0.061* | 0.011 | -0.008 | 0.024 | 0.087** | 0.192*** | 0.201*** | 0.124*** | -0.130*** | 1 | | |
| Edu | 0.107*** | -0.023 | 0.036 | 0.156*** | 0.021 | 0.139*** | 0.083** | 0.196*** | 0.143*** | -0.04 | 1 | |
| Post | 0.109*** | 0.103*** | -0.113*** | -0.011 | 0.180*** | 0.159*** | 0.003 | -0.167*** | 0.031 | 0.195*** | 0.045 | 1 |

*, **, and *** denote significance at the 10, 5, and 1% levels, respectively. (t-statistics are in parentheses).

### 4.3 Regression analysis

**4.3.1 CEO Narcissism and enterprise ambidextrous technological innovation.** Regression analysis was performed on the relevant variables according to models (1) and (2), respectively. This paper uses a panel model that controls years and industries to perform hierarchical regression. The core independent and dependent variables are first regressed, then the core independent and dependent variables are regressed after adding control variables. The regression results are shown in Table 5.

Columns (1) and (2) of Table 5 are the regression results of the core independent and dependent variables. Columns (3) and (4) are the regression results after adding the control variables at the CEO and enterprise levels and controlling for the fixed effects of industry and time. It can be seen that the regression coefficients of CEO narcissism are all positive and significant at the 1% level. This shows that CEO narcissism has a positive and significant effect on both exploratory and exploitative innovation, which verifies H1a and H1b. Moreover, the regression coefficients of CEO narcissism on exploratory innovation are larger than those on exploitative innovation, indicating that CEO narcissism has a greater positive impact on exploratory innovation than exploitative innovation. This result is consistent with the research conclusion Kashmiri (2017) put forward narcissistic managers can encourage enterprises to carry out innovation activities [9]. This paper effectively verifies that narcissistic CEOs can promote enterprises to carry out ambidextrous technological innovation in the Chinese context.

**Table 5. Benchmark regression results.**

| variable | Pat1 | Pat2 | Pat1 | Pat2 |
|---|---|---|---|---|
| | (1) | (2) | (3) | (4) |
| Nar | 2.627*** (0.389) | 2.562*** (0.416) | 1.334*** (0.329) | 1.042*** (0.392) |
| Debt | | | -0.527** (0.315) | 0.211 (0.376) |
| Size | | | 0.841*** (0.047) | 0.744*** (0.056) |
| Set | | | -0.005 (0.011) | -0.033** (0.014) |
| Pattern | | | 0.012 (0.118) | -0.018 (0.141) |
| Rd | | | 0.070* (0.009) | -0.009 (0.011) |
| Age | | | 0.007 (0.007) | -0.023*** (0.008) |
| Edu | | | 0.042 (0.043) | 0.004 (0.051) |
| Post | Post | Post | 0.065 (0.093) | 0.087 (0.111) |
| CONSTANT | 0.071 (0.839) | -1.07 (0.899) | -18.255*** (1.148) | -14.373*** (1.368) |
| Year | Yes | Yes | Yes | Yes |
| Industry | Yes | Yes | Yes | Yes |
| Observation | 798 | 798 | 796 | 796 |
| R-squared | 0.1319 | 0.2905 | 0.4729 | 0.4655 |
| Adjusted R-squared | 0.1152 | 0.2769 | 0.4571 | 0.4496 |

*, **, and *** denote significance at the 10, 5, and 1% levels, respectively. (t-statistics are in parentheses).

**4.3.2 The mediating role of CSR.** The mediating role of corporate social responsibility was tested using the test models (3)–(5). The regression results are shown in Table 6.

The regression results in column (1) show that when corporate social responsibility is used as the dependent variable, the coefficient of CEO narcissism is significantly positive at the 5% level, indicating that CEO narcissism will significantly promote corporate social responsibility. The results in columns (2)-(3) show that the increase in CEO narcissism significantly promotes both exploratory and exploitative innovation at the 1% level. The regression results in columns (4)-(5) show that CEO narcissism is significantly positive at the 1% and 10% levels for corporate exploratory and exploitative innovation after controlling for corporate social responsibility as an intermediary variable, respectively. Since the regression coefficient of CEO narcissism on CSR is significant and the regression coefficient of CSR on exploratory innovation is significant when CEO narcissism, CSR are regressed together on exploratory innovation, it can be concluded that there is an indirect effect. When CEO narcissism, CSR are regressed together on exploratory innovation, the regression coefficient of Nar on exploratory innovation is significant, indicating a significant direct effect. Finally, the regression coefficient of CEO narcissism on CSR is significantly the same sign as the regression coefficient of CSR on exploratory innovation when CEO narcissism and CSR are regressed together on exploratory innovation. This indicates a partial mediating effect. The evidence for a mediating effect of CSR in CEO narcissism and exploitative innovation is the same as described above.

**Table 6. The mediating role of CSR.**

| variable | CSR | Pat1 | Pat2 | Pat1 | Pat2 |
|---|---|---|---|---|---|
| | (1) | (2) | (3) | (4) | (5) |
| Nar | 8.403** | 1.333*** | 1.041*** | 1.279*** | 0.963* |
| | (3.401) | (0.329) | (0.392) | (0.330) | (0.393) |
| CSR | | | | 0.006* | 0.009* |
| | | | | (0.003) | (0.004) |
| Debt | -18.028*** | -0.527* | 0.211 | -0.410 | 0.380 |
| | (3.257) | (0.315) | (0.376) | (0.321) | (0.382) |
| Size | 5.120*** | 0.841*** | 0.744*** | 0.807*** | 0.695*** |
| | (0.482) | (0.047) | (0.056) | (0.050) | (0.060) |
| Set | 0.003 | -0.005 | -0.033** | -0.005 | -0.033** |
| | (0.118) | (0.011) | (0.014) | (0.011) | (0.014) |
| Pattern | 1.256 | 0.012 | 0.018 | 0.003 | -0.029 |
| | (1.222) | (0.118) | (0.141) | (0.118) | (0.141) |
| Rd | 0.039 | 0.070*** | -0.009 | 0.070*** | -0.009 |
| | (0.091) | (0.009) | (0.011) | (0.009) | (0.010) |
| Age | 0.093 | 0.007 | -0.023*** | 0.006 | -0.023*** |
| | (0.073) | (0.007) | (0.008) | (0.007) | (0.008) |
| Edu | 1.074** | 0.042 | 0.004 | 0.036 | -0.006 |
| | (0.440) | (0.043) | (0.051) | (0.043) | (0.051) |
| Post | -0.257 | 0.065 | 0.087 | 0.066 | 0.090 |
| | (0.965) | (0.093) | (0.111) | (0.093) | (0.111) |
| CONSTANT | -92.868*** | -18.255*** | -14.373*** | -17.654 | -13.504*** |
| | (11.858) | (1.148) | (1.368) | (1.191) | (1.418) |
| Year | Yes | Yes | Yes | Yes | Yes |
| Industry | Yes | Yes | Yes | Yes | Yes |
| Observation | 796 | 796 | 796 | 796 | 796 |
| R-squared | 0.3306 | 0.4729 | 0.4655 | 0.4752 | 0.4690 |
| Adjusted R-squared | 0.3107 | 0.4571 | 0.4496 | 0.4589 | 0.4525 |

*, **, and *** denote significance at the 10, 5, and 1% levels, respectively. (t-statistics are in parentheses).

The empirical results show that corporate social responsibility mediates the relationship between CEO narcissism and corporate exploratory and exploitative innovation.

**4.3.3 Robustness test.** The independent variable of CEO narcissism used in this paper is to standardize the three indicators it contains and take the average as the measurement standard. We conduct a principal component analysis of the three indicators and use the first main component to measure CEO narcissism (the eigenvalue is 2.27, and the variance contribution rate is 76%), then substitute it into the above model for testing. The benchmark regression results are shown in Table 7. The results of the mediation effect regression are shown in Table 8. The test indicates that CEO narcissism is significant for exploratory innovation at the 1% level and exploitative innovation at the 10% level. After adding corporate social responsibility as a mediating variable, CEO narcissism is significant for exploratory innovation at the 1% level and exploitative innovation at the 10% level.

Considering that the influence of CEO narcissism on enterprise ambidextrous innovation may have a lagging effect, this paper re-examines exploratory innovation and exploitative innovation with a lag of one period, respectively. The results of the benchmark regression are shown in Table 9, and the regression results of the mediation effect are shown in Table 10, which are not significantly different from the previous empirical results.

## 5 Conclusion

This paper takes China's Shanghai and Shenzhen A-share listed R&D-intensive companies as observation samples to explore the relationship between CEO narcissism and corporate

**Table 7. Principal component factor analysis—benchmark regression robustness test.**

| variable | Pat1 | Pat2 | Pat1 | Pat2 |
|---|---|---|---|---|
| | (1) | (2) | (3) | (4) |
| Nar | 0.242*** | 0.209*** | 0.121*** | 0.068* |
| | (0.037) | (0.040) | (0.031) | (0.037) |
| Debt | | | -0.539* | 0.219 |
| | | | (0.316) | (0.377) |
| Size | | | 0.849*** | 0.756*** |
| | | | (0.046) | (0.055) |
| Set | | | -0.005 | -0.031** |
| | | | (0.011) | (0.014) |
| Pattern | | | 0.010 | -0.006 |
| | | | (0.119) | (0.141) |
| Rd | | | 0.070*** | -0.010 |
| | | | (0.009) | (0.011) |
| Age | | | 0.006 | -0.024*** |
| | | | (0.007) | (0.008) |
| Edu | | | 0.036 | -0.003 |
| | | | (0.043) | (0.051) |
| Post | | | 0.046 | 0.070 |
| | | | (0.093) | (0.111) |
| CONSTANT | 0.677 | 0.486 | -18.053*** | -14.324*** |
| | (0.836) | (0.899) | (-1.154) | (1.377) |
| Year | Yes | Yes | Yes | Yes |
| Industry | Yes | Yes | Yes | Yes |
| Observation | 798 | 798 | 796 | 796 |
| R-squared | 0.1287 | 0.2813 | 0.4721 | 0.4630 |
| Adjusted R-squared | 0.1120 | 0.2676 | 0.4563 | 0.4470 |

*, **, and *** denote significance at the 10, 5, and 1% levels, respectively. (t-statistics are in parentheses).

**Table 8. Principal component factor analysis—robustness test of mediation effects.**

| Variable | CSR | Pat1 | Pat2 | Pat1 | Pat2 |
|---|---|---|---|---|---|
| | (1) | (2) | (3) | (4) | (5) |
| Nar | 0.708** | 0.121*** | 0.068* | 0.116*** | 0.061* |
| | (0.320) | (0.031) | (0.037) | (0.031) | (0.037) |
| CSR | | | | 0.007* | 0.010** |
| | | | | (0.003) | (0.004) |
| Debt | -18.063*** | -0.538* | 0.219 | -0.418 | 0.394 |
| | (3.261) | (0.316) | (0.377) | (0.321) | (0.383) |
| Size | 5.260*** | 0.849*** | 0.756*** | 0.814*** | 0.705*** |
| | (0.480) | (0.046) | (0.055) | (0.050) | (0.059) |
| Set | 0.008 | -0.005 | -0.031** | -0.005 | -0.031** |
| | (0.118) | (0.011) | (0.014) | (0.011) | (0.014) |
| Pattern | 1.274 | 0.010 | -0.006 | 0.002 | -0.018 |
| | (1.224) | (0.118) | (0.141) | (0.118) | (0.141) |
| Rd | 0.032 | 0.069*** | -0.010 | 0.068*** | -0.010 |
| | (0.091) | (0.009) | (0.011) | (0.009) | (0.010) |
| Age | 0.090 | 0.006 | -0.024*** | 0.006 | -0.024*** |
| | (0.073) | (0.007) | (0.008) | (0.007) | (0.008) |
| Edu | 1.029** | 0.037 | -0.003 | 0.029 | -0.013 |
| | (0.439) | (0.043) | (0.051) | (0.043) | (0.051) |
| Post | -0.382 | 0.046 | 0.070 | 0.048 | 0.073 |
| | (0.962) | (0.093) | (0.111) | (0.093) | (0.111) |
| CONSTANT | -91.815*** | -18.053*** | -14.324*** | —17.444*** | -13.433*** |
| | (11.918) | (1.154) | (1.377) | (1.196) | (1.425) |
| Year | Yes | Yes | Yes | Yes | Yes |
| Industry | Yes | Yes | Yes | Yes | Yes |
| Observation | 796 | 796 | 796 | 796 | 796 |
| R-squared | 0.3296 | 0.4721 | 0.4630 | 0.4745 | 0.4668 |
| Adjusted R-squared | 0.3096 | 0.4563 | 0.4470 | 0.4582 | 0.4502 |

*, **, and *** denote significance at the 10, 5, and 1% levels, respectively. (t-statistics are in parentheses).

ambidextrous technological innovation and the mediating role of corporate social responsibility in the relationship. The research results show that: (1) CEO narcissism has a significant positive impact on both exploratory and exploitative innovation. Compared with exploitative innovation, CEO narcissism has a greater positive impact on exploratory innovation; (2) CSR has a mediating role between CEO narcissism and corporate exploratory innovation; (3) CSR has a mediating role between CEO narcissism and corporate exploitative innovation.

Analyzing the empirical results, we believe that the reasons may be: (1) Narcissistic CEOs have risk-taking tendencies and higher psychological expectations of risk-taking, so they will increase the company's risky expenditures, such as increasing R&D investment and promoting the company's innovative output, and many more. In addition, narcissistic CEOs are more likely to have alliances, mergers and acquisitions awareness, and the possibility of launching new technology research and development projects, which will drive the executive team to increase their attention to the progress of technological innovation. Therefore, CEO narcissism can promote the development of ambidextrous technological innovation in enterprises;(2) Exploratory innovation is a breakthrough innovation activity with high risk, and its results will become the core competitive advantage of enterprises, thereby helping enterprises to occupy a dominant position in the industry. Narcissistic CEOs are more inclined to choose strategic decisions that maximize personal fulfillment and are more likely to attract attention and

**Table 9. One period lag—benchmark regression robustness test.**

| variable | Pat1 | Pat2 | Pat1 | Pat2 |
|---|---|---|---|---|
| | (1) | (2) | (3) | (4) |
| Nar | 2.161*** | 1.823*** | 0.142*** | 0.982*** |
| | (0.325) | (0.373) | (0.295) | (0.355) |
| Debt | | | -0.362 | -0.293 |
| | | | (0.225) | (0.270) |
| Size | | | 0.662*** | 0.671*** |
| | | | (0.035) | (0.042) |
| Set | | | -0.021*** | -0.028*** |
| | | | (0.007) | (0.008) |
| Pattern | | | 0.040*** | 0.088 |
| | | | (0.086) | (0.103) |
| Rd | | | 0.045*** | -0.001 |
| | | | (0.006) | (0.007) |
| Age | | | -0.003 | -0.029*** |
| | | | (0.005) | (0.006) |
| Edu | | | 0.053* | -0.066* |
| | | | (0.028) | (0.034) |
| Post | | | 0.120** | 0.258*** |
| | | | (0.066) | (0.080) |
| CONSTANT | 0.677 | -0.227 | -14.388*** | -13.271*** |
| | (0.836) | (0.850) | (0.971) | (1.169) |
| Year | Yes | Yes | Yes | Yes |
| Industry | Yes | Yes | Yes | Yes |
| Observation | 798 | 798 | 796 | 796 |
| R-squared | 0.0991 | 0.2585 | 0.3462 | 0.4072 |
| Adjusted R-squared | 0.0879 | 0.2492 | 0.3342 | 0.3963 |

*, **, and *** denote significance at the 10, 5, and 1% levels, respectively. (t-statistics are in parentheses).

positive comments from others. Therefore, exploratory innovation is more in line with the more aggressive and bold strategic decisions that narcissistic CEOs tend to make. And at present, most of the enterprises in China still focus on "improved" utilization innovation as the main research object, so the influence of narcissism on exploratory innovation may be greater. In addition, in the empirical results, the regression coefficient of enterprise size is significantly positive, indicating that the innovation results of large-scale enterprises are better, for the development of technological innovation activities of enterprises requires certain resource accumulation and background support.

In addition, narcissistic CEOs may promote corporate social responsibility because they make bold decisions and tend to overestimate returns on investment, so they may decide to devote more resources to CSR. And fulfilling social responsibilities is conducive to establishing a good social image of the company and improving the CEO's reputation, which is in line with the narcissistic CEO's desire to pursue fame, fortune, and attention. As a result, fulfilling corporate social responsibility will further bring more resources to the enterprise, such as employees, customers, business partners, suppliers, and other internal and external stakeholders. So as to promote the dual technological innovation of enterprises.

From the perspective of CEO personality traits and corporate innovation heterogeneity, this paper studies the impact of CEO narcissism on corporate technological innovation, enriching the research on upper echelon theory. It also discusses the measurement indicators and methods of CEO narcissism in the Chinese context and incorporates corporate social

**Table 10. One period of lag—a robustness test of the mediation effect.**

| variable | CSR | Pat1 | Pat2 | Pat1 | Pat2 |
|---|---|---|---|---|---|
| | (1) | (2) | (3) | (4) | (5) |
| Nar | 11.903*** | 0.142*** | 0.982*** | 1.079*** | 0.836** |
| | (2.733) | (0.295) | (0.355) | (0.297) | (0.360) |
| CSR | / | / | / | 0.005* | 0.012*** |
| | | | | (0.003) | (0.004) |
| Debt | -17.790*** | -0.362 | -0.293 | -0.268 | -0.074 |
| | (2.081) | (0.224) | (0.270) | (0.230) | (0.276) |
| Size | 3.793*** | 0.662*** | 0.671*** | 0.642*** | 0.624*** |
| | (0.320) | (0.035) | (0.042) | (0.036) | (0.043) |
| Set | -0.139** | -0.021*** | -0.028*** | -0.021*** | -0.027*** |
| | (0.061) | (0.007) | (0.008) | (0.007) | (0.008) |
| Pattern | 1.495* | 0.340*** | -0.088 | 0.332*** | 0.070 |
| | (0.795) | (0.086) | (0.103) | (0.086) | (0.103) |
| Rd | -0.180*** | 0.045*** | -0.001 | 0.046*** | 0.001 |
| | (0.055) | (0.006) | (0.007) | (0.006) | (0.007) |
| Age | 0.065 | -0.003 | -0.028*** | -0.003 | -0.029*** |
| | (0.047) | (0.005) | (0.006) | (0.005) | (0.006) |
| Edu | 0.561** | 0.053* | 0.066* | 0.050* | 0.059* |
| | (0.260) | (0.028) | (0.034) | (0.028) | (0.051) |
| Post | -0.165 | 0.120* | 0.258** | 0.119* | 0.256*** |
| | (0.615) | (0.066) | (0.080) | (0.066) | (0.080) |
| CONSTANT | -54.484*** | -14.389*** | -13.271*** | —14.102*** | -12.600*** |
| | (8.997) | (0.971) | (1.169) | (0.983) | (1.179) |
| Year | Yes | Yes | Yes | Yes | Yes |
| Industry | Yes | Yes | Yes | Yes | Yes |
| Observation | 796 | 796 | 796 | 796 | 796 |
| R-squared | 0.2611 | 0.3462 | 0.4072 | 0.3478 | 0.4126 |
| Adjusted R-squared | 0.2475 | 0.3342 | 0.3963 | 0.3353 | 0.4013 |

*, **, and *** denote significance at the 10, 5, and 1% levels, respectively. (t-statistics are in parentheses).

responsibility into the discussion framework. The research results help companies formulate CEO appointment standards, formulate long-term innovation-driven strategies, and provide Chinese enterprises with important practical significance to achieve high-quality development.

## 6 Implications and limitation

Enterprise is the main body of innovation activities, and the CEO, as the core figure of enterprise decision-making, its psychological characteristics will have an important impact on enterprise innovation. In order to make the CEO take the responsibility of promoting the innovation and development of the enterprise more effectively, the positive and negative effects of the CEO's narcissistic traits on the enterprise should be viewed comprehensively and objectively. Although some studies suggest that CEO narcissism will make companies face greater investment risks, the overestimation of personal ability and high level of risk-taking by CEO narcissism can also promote corporate innovation activities. In the context of China's construction as an innovative country, the conclusions of this paper have important theoretical reference value for companies to select CEOs. Companies should evaluate candidates from multiple dimensions of personal ability and comprehensive quality and pay attention to the inspection of the CEO's psychological characteristics. When an enterprise is in a bottleneck

period of innovation or cross-field development, narcissistic candidates can be boldly selected to lead the enterprise to achieve breakthroughs in new fields. From the CEO's perspective, narcissistic CEOs should establish a comprehensive understanding of themselves, clarify the possible role of narcissism in corporate decision-making, and control the scope of risks while promoting corporate innovation strategies. In addition, if an enterprise wants to achieve high-quality development, it needs to invest resources to realize its responsibility and contribution to the enterprise and society. Currently, many enterprises still ignore the urgency and necessity of fulfilling social responsibility and blindly pursue the development of individual enterprises. The fulfillment of social responsibility plays an important role in the long-term development of enterprises and social construction. Although there is a certain resource competition relationship between the fulfillment of corporate social responsibility and the development of technological innovation, proper fulfillment of corporate social responsibility can obtain employee enthusiasm, customer support, and government policy support from corporate employees, consumers, partners, and other stakeholders. External resources, and ultimately realize the long-term development of the enterprise. Therefore, companies should encourage narcissistic CEOs to make decisions related to fulfilling corporate social responsibility and maintaining the relationship between the company and stakeholders. And formulate complementary strategies to balance the contradiction of resource consumption between social responsibility and technological innovation and achieve high-quality, sustainable development.

The research described in this paper may have several shortcomings: Since the sample comes from R&D-intensive listed companies, although the influence of industry variables is considered, there are still certain limitations in selecting enterprise types and industries. Therefore, future analyses can be performed for other business types and industries. In addition, this paper selects non-interventional indicators as the three dimensions to measure CEO narcissism and obtains the final data through the manual collection. In future research, a more detailed dimension division can be carried out, and a more accurate measurement method can be used.

## Supporting information

**S1 Data. Raw data for data analysis.**
(XLSX)

## Author Contributions

**Conceptualization:** Xingqiu Hu, Feifei Yu.

**Data curation:** Zeping Wang.

**Formal analysis:** Zeping Wang, Feifei Yu.

**Funding acquisition:** Xingqiu Hu.

**Investigation:** Zeping Wang, Feifei Yu.

**Methodology:** Zeping Wang, Feifei Yu.

**Project administration:** Xingqiu Hu, Feifei Yu.

**Software:** Zeping Wang.

**Supervision:** Zeping Wang.

**Validation:** Xingqiu Hu.

**Writing – original draft:** Zeping Wang.

**Writing – review & editing:** Zeping Wang, Xingqiu Hu, Feifei Yu.

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
