## [Decision Letter · Decision Letter 0]

6 Nov 2022

PONE-D-22-28052How does CEO narcissism affect enterprise Ambidextrous Technological Innovation? The mediating role of corporate social responsibilityPLOS ONE

Dear Dr. Wang,

Thank you for submitting your manuscript to PLOS ONE. After careful consideration, we feel that it has merit but does not fully meet PLOS ONE’s publication criteria as it currently stands. Therefore, we invite you to submit a revised version of the manuscript that addresses the points raised during the review process.

We look forward to receiving your revised manuscript.

Kind regards,

Claudia Noemi González Brambila, Ph.D.

Academic Editor

PLOS ONE

3. Please remove your figures from within your manuscript file, leaving only the individual TIFF/EPS image files, uploaded separately. These will be automatically included in the reviewers’ PDF.

Reviewers' comments:

Reviewer's Responses to Questions

**Comments to the Author**

1. Is the manuscript technically sound, and do the data support the conclusions?

Reviewer #1: Yes

Reviewer #2: Yes

Reviewer #3: Yes

2. Has the statistical analysis been performed appropriately and rigorously? 

Reviewer #1: Yes

Reviewer #2: Yes

Reviewer #3: Yes

3. Have the authors made all data underlying the findings in their manuscript fully available?

Reviewer #1: Yes

Reviewer #2: Yes

Reviewer #3: Yes

4. Is the manuscript presented in an intelligible fashion and written in standard English?

Reviewer #1: Yes

Reviewer #2: Yes

Reviewer #3: Yes

5. Review Comments to the Author

Reviewer #1: Dear Authors: I enjoyed reading your manuscript. Below my concerns:

1) Abstract is long and confusing

2) Research gap is unclear. You need to slowly bring the reader to your stated gap

3) Research question needs to be clarified and simplified

4) Clearly connect your research gap with your research question

5) Rationale for bringing CSR into your research needs to be better explained

6) Your hypothesis do not seem to be theoretically supported. You mention different theories throughout the paper but it is not clear which ones you are building on and why

7) The mediating mechanism logic of CSR in your hypothesis needs to be explained better

8) Need to provide more details on what is China hexus

I wish you the best of luck,

Reviewer #2: Overall great job and well written paper.

In order to improve the quality, I would suggest the following:

1. include a separate literature review section to discuss the existing state of literature and better position your research.

2. most of the citations are outdated, it is recommended to include most recent studies.

3. In the methodology section, reliability and validity statistics are missing for measurement scale. Please provide these statistics before testing the hypotheses.

Reviewer #3: the paper was based on the CEO narcissism, ambidextrous technological innovation, CSR and regression analysis. it made use of upper echelons theory and others. the paper is unique due to the psychology and personality traits used rather than the common demographic characteristics. the paper made few statements requiring references such as in the latter part of introduction where the author highlighted the gaps filled. there was no special information to ascertain that the measures of a construct was actually meant for it (content and construct validity). this is true for CEO narcissism, the mediator CSR and the ambidextrous technological innovation. also, conducting separate regression for variables hierarchically leads to issue of confounding. the mediation results did not specify the type of mediation discovered. Finding the average to replace CEO narcissism construct is not too good because the variable is losing importance such as in Confounding problem. it is noteworthy tp state that the references follow no order and that reduces the quality of the paper.

6. PLOS authors have the option to publish the peer review history of their article (what does this mean?). If published, this will include your full peer review and any attached files.

---

## [Author Response · Author response to Decision Letter 0]

6 Dec 2022

Dear Editor and Reviewer,

Thanks for your comments and we are delighted to have the opportunity to revise our manuscript “How does CEO narcissism affect enterprise Ambidextrous Technological Innovation? The mediating role of corporate social responsibility” (PONE-D-22-28052). We have modified the manuscript according to the suggestions. Revised parts are marked in red in the revised manuscript. Below are our point-to-point responses. If there are any other modifications we could make, we would like very much to modify them and we really appreciate your help.

Sincerely yours,

Zeping Wang, on behalf of all co-authors.

 

COMMENTS FROM THE EDITOR:

Response: Thank you for your reminder. We have revised the formatting of the manuscript and the naming of the files.

Response: Thank you for your reminder. When resubmitting, we will fill out the "Funding Information" section correctly.

3. Please remove your figures from within your manuscript file, leaving only the individual TIFF/EPS image files, uploaded separately. These will be automatically included in the reviewers’ PDF.

Response: Thank you for your guidance. We have removed the figures from the file and uploaded the TIFF image file.

 

Reviewer #1:

I enjoyed reading your manuscript. Below my concerns:

1. Abstract is long and confusing.

Response: Thank you for your precise suggestions. We have revised the abstract as you suggested, sorting out the logic and refining the language expression (Pages 1-2, lines 9-22). 

2. Research gap is unclear. You need to slowly bring the reader to your stated gap. 

Response: Thank you for your useful suggestions. We have polished the introduction section for the introduction of the literature on research gaps, describing exactly the research gaps filled in this paper (Pages 4-5, lines 64-92).

3. Research question needs to be clarified and simplified.

Response: Thank you for the points you have made. We have simplified the description of the research questions to present more clearly the content and significance of the article under study (Page 5, lines 93-98).

4. Clearly connect your research gap with your research question.

Response: Thank you for your pertinent suggestions, and we have made corrections in accordance with your suggestions. We add an explanation of the research gap in the second half of the introduction and present the research questions of this paper based on the research gap (Pages 4-5, lines 64-98).

5. Rationale for bringing CSR into your research needs to be better explained

Response: Thank you for your suggestions. We add in the introduction section why it is important to study the way the independent variable affects the dependent variable and explain why CSR is chosen as a mediating variable (Pages 4-5, lines 72-92).

6. Your hypothesis do not seem to be theoretically supported. You mention different theories throughout the paper but it is not clear which ones you are building on and why.

Response: Thank you for your pertinent questions. We have added an analysis of upper echelon theory in the theory and hypothesis section and explained how it relates to the research questions in this paper (Pages 6-7, lines 121-135).

7. The mediating mechanism logic of CSR in your hypothesis needs to be explained better.

Response: Thank you for your thoughtful suggestions. We have added references to CSR mediating mechanisms in the Theory and Hypotheses section. More specifically, we discuss how CSR plays a role between CEO narcissism and dual corporate technology innovation (Pages 10-12, lines 209-235; Pages 12-13, lines 249-261)

8. Need to provide more details on what is China hexus.

Response: Thank you for your suggestion, and we apologize that we overlooked it. We have added an explanation about China hexus and the reasons for choosing it (Page 20, lines 382-387). 

“CSR data is obtained from Hexun Station‘s 2015-2020 CSR report rating database. Hexun Station is a more authoritative website that publishes comprehensive CSR scores of Chinese listed companies. Hexun Station publishes a comprehensive social responsibility rating of Chinese listed companies every year as a measure of CSR performance. It can reflect the social responsibility performance of enterprises in an objective manner.”

Reviewer #2:

Overall great job and well written paper.

In order to improve the quality, I would suggest the following:

1. include a separate literature review section to discuss the existing state of literature and better position your research.

Response: Thank you for your important suggestions. We have added a compilation of references in the introduction section, which discusses the state of the existing literature and leads to the question to be studied in the article (Pages 3-4, lines 43-63).

2. most of the citations are outdated, it is recommended to include most recent studies.

Response: Thanks to your suggestion, we read the latest relevant research and added it to the article, such as reference 31 (page 11, line 214), reference 43 (page 15, line 310), reference 44 (page 16, line 320) etc.

3. In the methodology section, reliability and validity statistics are missing for measurement scale. Please provide these statistics before testing the hypotheses.

Response: Thank you for your comment and we have revised as your suggestion. In order to guarantee the reliability and validity of the data, we have done a reliability test on the scale. The details are added in page 19, lines 370-374.

Reviewer #3:

the paper was based on the CEO narcissism, ambidextrous technological innovation, CSR and regression analysis. it made use of upper echelons theory and others. the paper is unique due to the psychology and personality traits used rather than the common demographic characteristics. 

1. the paper made few statements requiring references such as in the latter part of introduction where the author highlighted the gaps filled. 

Response: Thank you for your proposal. We have added a compilation and analysis of relevant references in the introduction section. By combing them, we clarify the research gaps in this paper and provide references to support the questions studied in this paper (Pages 3-5, lines 43-92).

2. There was no special information to ascertain that the measures of a construct was actually meant for it (content and construct validity). this is true for CEO narcissism, the mediator CSR and the ambidextrous technological innovation. 

Response:Thank you for your valid guidance. To demonstrate the content and construct validity of the measurement aspects of this study, we have added explanations of the main variables in the Definition and Measurement of Variables section, as well as further analysis of why the indicators in the text were chosen to measure the variables (Pages 14-15, lines 296-310). Additional tests of reliability and validity were added for the scales of the independent variables (Page 19, lines 370-374).

3. Also, conducting separate regression for variables hierarchically leads to issue of confounding. the mediation results did not specify the type of mediation discovered. 

Response: Thank you for your pertinent suggestions. In section 4.3.2 The mediating role of CSR, we explain in detail the model used and the results of the data analysis, which show that the mediating effect is significant. And we also distinguish the types of mediating effects in this study (Pages 27-28, lines 473-483).

4. Finding the average to replace CEO narcissism construct is not too good because the variable is losing importance such as in Confounding problem. 

Response: Thank you for your suggestions. We added a reliability test for the indicators of the independent variable scale and added references related to taking the mean of the indicators as variable data (Page 19, lines 370-374). In the Robustness Test section, we replaced the method of taking the mean with the method of principal component factor analysis, processed the three indicators of CEO narcissism again, synthesized the CEO narcissism data, and re-performed the experiment with this data. The experimental results remained consistent with the results of replacing CEO narcissism with the mean (Page 28, lines 488-498).

5. it is noteworthy to state that the references follow no order and that reduces the quality of the paper. 

Response: Thank you for your help. We are very sorry for our carelessness and have reorganized the references.

---

## [Decision Letter · Decision Letter 1]

8 Jan 2023

How does CEO narcissism affect enterprise Ambidextrous Technological Innovation? The mediating role of corporate social responsibility

PONE-D-22-28052R1

Dear Dr. Wang,

We’re pleased to inform you that your manuscript has been judged scientifically suitable for publication and will be formally accepted for publication once it meets all outstanding technical requirements.

Kind regards,

Claudia Noemi González Brambila, Ph.D.

Academic Editor

PLOS ONE

Additional Editor Comments (optional):

Reviewers' comments:

Reviewer's Responses to Questions

**Comments to the Author**

1. If the authors have adequately addressed your comments raised in a previous round of review and you feel that this manuscript is now acceptable for publication, you may indicate that here to bypass the “Comments to the Author” section, enter your conflict of interest statement in the “Confidential to Editor” section, and submit your "Accept" recommendation.

Reviewer #1: All comments have been addressed

Reviewer #2: All comments have been addressed

2. Is the manuscript technically sound, and do the data support the conclusions?

Reviewer #1: Yes

Reviewer #2: Yes

3. Has the statistical analysis been performed appropriately and rigorously? 

Reviewer #1: Yes

Reviewer #2: Yes

4. Have the authors made all data underlying the findings in their manuscript fully available?

Reviewer #1: Yes

Reviewer #2: Yes

5. Is the manuscript presented in an intelligible fashion and written in standard English?

Reviewer #1: Yes

Reviewer #2: Yes

6. Review Comments to the Author

Reviewer #1: Congratulations. I see that you have properly addressed my concerns. I wish you the best in pursuing your future research endeavors,

Reviewer #2: (No Response)

7. PLOS authors have the option to publish the peer review history of their article (what does this mean?). If published, this will include your full peer review and any attached files.

Reviewer #1: No

Reviewer #2: **Yes: **

---

## [Editor Report · Acceptance letter]

11 Jan 2023

PONE-D-22-28052R1 

How does CEO narcissism affect enterprise Ambidextrous Technological Innovation? The mediating role of corporate social responsibility 

Dear Dr. Wang:

I'm pleased to inform you that your manuscript has been deemed suitable for publication in PLOS ONE. Congratulations! Your manuscript is now with our production department. 

Kind regards, 

on behalf of

Dr. Claudia Noemi González Brambila 

Academic Editor

PLOS ONE